# Assessing the Use of Social Cognitive Theory Components in Cooking and Food Skills Interventions

**DOI:** 10.3390/nu15051287

**Published:** 2023-03-04

**Authors:** Paola Gordillo, Melissa Pflugh Prescott

**Affiliations:** Department of Food Science and Human Nutrition, University of Illinois Urbana Champaign, Urbana, IL 61801, USA

**Keywords:** adult, social cognitive theory, cooking, self-efficacy, behavior

## Abstract

Increased cooking skill development may reduce the risk of disease and promote healthy eating behaviors in the home. The social cognitive theory (SCT) is one of the most common theories used in cooking and food skill interventions. This narrative review aims to understand how commonly each SCT component is implemented in cooking interventions, as well as identifying which components are associated with positive outcomes. The literature review was conducted using three databases: PubMed, Web of Science (FSTA and CAB), and CINHAL, yielding thirteen included research articles. None of the studies in this review comprehensively included all SCT components; at most, five of the seven were defined. The most prevalent SCT components were behavioral capability, self-efficacy, and observational learning, and the least implemented component was expectations. All studies included in this review yielded positive outcomes for cooking self-efficacy and frequency, except for two studies with null outcomes. Findings from this review suggest that the SCT may not be fully realized, and future studies should continue to define how theory influences intervention design for adult cooking interventions.

## 1. Introduction

Over the past several decades, Americans have grown to rely on the convenience of purchasing prepared foods away from the home [1]. Ultra-processed, ready-made foods are characteristically energy-dense and have become dominant across high and middle-income countries [2]. Hall et al. (2019) established causal evidence linking ultra-processed diets to increased caloric intake and weight gain [3]. Monsivais et al. (2014) suggest that spending <1 h/day on food preparation was associated with more money spent on food away from the home and more frequent use of fast food restaurants compared to those who spent more time cooking [4]. Yet, researchers suggests a loss of the necessary skills required to prepare a meal from scratch [5]. Meal preparation encompasses a series of complex daily living activities that includes commuting to the grocery store, shopping for ingredients, managing money, preparing and choosing a recipe, and cooking a meal [6]. The term ‘cooking skills’ within public health nutrition has been used to describe a combination of mechanical and physical food preparation skills used in the home, such as chopping vegetables or cooking rice [7]. 

Hollywood et al. (2018) conducted a literature review to connect cooking and food skills interventions with their behavior change techniques and theoretical underpinnings [8]. In this critical review, the (SCT) was the most common behavior change theory used, appearing in nine of fourteen included studies. The SCT is defined by Bandura’s reciprocal model (reciprocal determinism) in which personal factors, environmental influences, and behavior continually interact, and people learn not only by their actions but by observing the actions of others and the results of those actions [9]. 

The SCT consists of seven components that facilitate reciprocal determinism and, ultimately, behavior change: *Self-efficacy* is the conviction that one can successfully execute a behavior to produce specific outcomes [10], such as feeling confident that one can select, purchase, and prepare vegetables to improve dietary intake. The *environment* describes the social (e.g., family, peers), physical (e.g., weather, location) and economic (e.g., product availability, price changes) external factors influencing behavior. Providing participants with complimentary fruit and vegetable take-home bags would be an environmental change because the accessibility of the food is increased. An environment change can be impactful, but a person must know how to execute the desired behavior change. *Behavioral capability* refers to a person’s actual ability to perform a behavior themselves, like one’s ability to cook seasonal vegetables. Observing another person modeling a specific behavior, also known as *observational learning*, can make it easier to reproduce desired behavior, such as having participants watch a cooking instructor prepare vegetables before they attempt it themselves. *Expectations* portray anticipated outcomes from executing a behavior change, such as increasing vegetable variety during cooking practices to expand vegetable preferences. *Self-regulation* facilitates people’s ability to control their behavior. Through this, they may create, alter, and commit to goals to achieve a desired outcome, such as setting a goal to purchase two new vegetables a week to increase availability in the home. When accomplishing a behavior change, *reinforcements* can be shown through incentives or other recognitions of success, such as winning a prize for effectively using a seasonal vegetable in a main dish. 

While the SCT has been established as the most dominant behavior theory used in evidence-based practice cooking and food skills interventions, Hollywood et al. (2018) did not explore the extent to which the SCT was applied in these interventions. It is unclear whether cooking interventions based on the SCT incorporated all seven components of the theory or only implemented a fraction of them. To address this gap, the objective of the present study is to assess how commonly each SCT component is implemented in cooking interventions, as well as which components are associated with the most positive outcomes. 

## 2. Materials and Methods

In this critical review, three databases, PubMed, Web of Science (Food Science & Technology Abstracts (FSTA) and Commonwealth Agricultural Bureaux (CAB)), and CINHAL were used to search for peer-reviewed journal articles with the following search string: (Adult* OR “Health Coach*” OR “Peer Educator*” OR “Motivational coach*” OR “Motivation Coach” OR “well* coach*” OR “Community Health Worker*”) AND (“Goal* Setting*” OR “Behavior* Counsel*” OR “Barrier* to change*” OR Reward* OR Reinforce* OR “Self efficacy*” OR “Patient Centered Care*” OR “Social Cognitive Theory*” OR Motivation* OR Behavioral Capability* OR “Follow up prompt*” OR “Intention* to Chang*” OR “Behavior* Influence” OR “Person* Influence*” OR “Health* Behavior* Management*” OR “Behavior Change Technique*”) AND (“cook* skill*” OR “Home cook*” OR “Processed Food*” OR “Eat at home*” OR “Cook* Attitude*” OR “Cooking frequen*” OR “cooking demonstration*” OR “Improv* cook*” OR “Cook* Knowledge*” OR “eating in” OR “Culinary* Improv*” OR Cook* OR “Cook* Confidence*” OR “Cooking Competence*”). Articles from the search string were imported into Rayyan, and a total of 802 articles were identified in the search of all three databases. Two researchers independently screened the abstracts to determine whether they examined interventions related to cooking behaviors and met other inclusion/exclusion criteria (see Table 1). Articles that did not provide enough information in the abstract were reviewed in full. Articles were then exported into Zotero, and duplicates were removed. There were no exclusion criteria related to publication date or study design. If an intervention targeted children, the results had to include stratification between adults and children to allow for the isolation of results. 

## 3. Results

### 3.1. Study Characteristics

#### Publication, Population, Study Sample, Study Design

Thirteen studies were included in this review. Nine studies were conducted in the United States of America [11,12,13,14,15,16,17,18,19] and one each in Australia [20], Canada [21], Ecuador [22], and Ireland [23]. The sample size ranged from 54 to 336. While some studies included child family members, this review focuses on the results for adult participants. Six interventions involved caregivers of children, ranging from infants to 12 years of age [12,14,17,18,19,22], and one was exclusive to mothers [22]. The most frequent study design (*n* = 5) was a non-controlled trial [11,12,14,16,20]. Non-randomized controlled trials (*n* = 3) [17,21,22], randomized controlled trials (*n* = 3) [15,18,19], cluster randomized controlled trial (*n* = 1) [13], and parallel randomized controlled trial (*n* = 1) [23] were also included. Cooking self-efficacy [12,13,14,16,17,20,23] and frequency [11,12,13,14,19,21,22] were the only two cooking outcomes measured in the 13 included papers. Some of the studies simultaneously assessed outcomes related to cooking, such as attitudes or self-efficacy for consuming fruits and vegetables [11,20,23], access to fruits and vegetables [13,20], and fruit and vegetable serving frequency [11,15]. 

### 3.2. Intervention Characteristics

#### Mode of Delivery, Study Duration, Intervention Providers

All interventions were face-to-face format except for one study [15] where participants met in-person but watched a four-episode cooking show intervention on how college students can prepare, shop, and plan meals. Study duration ranged from 4 weeks to 12 months long. The interventions were delivered by volunteer students and/or community members [20,21,22,23], extension staff [14,19], nutrition educators [12,17], trained facilitators or staff [11,14], an investigator [11], a professional chef [12], and a registered dietitian [15]. Three articles did not specify the providers of the intervention [13,16,18]. A summary table of the included studies’ characteristics and key outcomes are found in Table 2.

### 3.3. Social Cognitive Theory (SCT) Components

Table 3 outlines how the SCT was applied in each included article. None of the articles reported using all the SCT components. At the upper range, seven articles defined how five out of the seven SCT components were used in their intervention [12,13,14,19,20,22,23]. At the lower range, three articles defined just three of the components [11,15,16]. All articles defined how behavioral capability, self-efficacy, and observational learning were used in their intervention [11,12,13,14,15,16,17,18,19,20,21,22,23]. The least used SCT component, expectations, was only defined in one study [13]. In combination with the SCT, some studies used the socio-ecological model [13,18,20,21], 4-H experiential model [14,19], health belief model [22], and the stages of change model [16] as a framework for their study. The remainder of the results section was organized to include how each study defined individual SCT components. Results for each study were given the first time the author is mentioned, but relevant study outcomes were also highlighted in Table 2.

#### 3.3.1. Environment

Eight studies [12,13,14,17,20,21,22,23] had interventions with a defined environmental component. Leone et al. (2018) improved produce availability, accessibility, affordability, and accommodations to change the food environment and people’s perceptions of it [13]. Leone et al. (2018) increased participant self-efficacy for incorporating more F&V into snacks (*p* = 0.02), making up a vegetable dish with what they had on hand (*p* = 0.03), and cooking vegetables in a way that is appealing to the family (*p* = 0.048). Similarly, Brimblecombe et al.’s (2018) study used a price discount strategy to increase affordability for participants; results indicated self-efficacy to cook and try new vegetables did not differ between baseline and end of intervention, but was lower 24 weeks post-intervention [20]. Mead et al. (2013) collaborated with local food stores, retailers, and other partners to increase the availability and accessibility of healthier food options [21]. The intervention participants in Mead et al. (2013) significantly reduced their average frequency of unhealthy food acquisition relative to the comparison group (change = −5.28, *p* = 0.0019) but had no significant improvements in food preparation scores. Overcash et al. (2018) and Pooler et al. (2017) increased availability by providing the food for participants directly [12,17]. Overcash et al. (2018) yielded increased parental cooking confidence (4.0 to 4.4/5.0) and healthy food preparation (3.6 to 3.9/5.0) [12]. The participants in Pooler et al. (2017) increased food resource management (FRM) practices (*p* = 0.002) and confidence (*p* < 0.001) [17]. Of the five studies that addressed the environment component by making food more accessible or available, four studies had positive outcomes including increased home availability of foods [12,17], increased affordability of vegetables [20], and decreased unhealthy food acquisition [21].

Three studies positively changed the social environment of participants by offering a support system to facilitate learning how to cook and improving mealtime dynamics. McHugh Power et al.’s (2016) study, in which a volunteer prepared and shared a meal with a participant at risk of social isolation, resulted in increased self-efficacy in the treatment group compared with the control (*p* = 0.054) [23]. Roche et al.’s (2017) study offered cooking clubs that provided social support for mothers to cook traditional food items, as well as an optional knitting club to discuss health issues and nutrition topics [22]. Roche et al.’s (2017) frequency to consume leafy greens, such as stinging nettle, resulted in 74% (*n* = 119) using it in a dish versus 21% in comparison communities. Miller et al.’s (2016) intervention had an extensive support system to improve culinary competence, food purchasing, and mealtime behaviors, which resulted in decreased procurement of fast food evening meals 1–2 days per week by 76% to 54% (*p* = 0.033 at post-test) and increased (16% to 40%) results for no fast food evening meals [14]. Miller et al.’s 2016 study also increased cooking skill confidence from 75% to 86% (*p* = 0.015) [14]. All three studies that addressed the (social) environment component had positive outcomes relating to self-efficacy and mealtime enjoyment [23], increased frequency to consume leafy greens [22], increased cooking skills confidence, and decreased consumption of fast food evening meals [14].

#### 3.3.2. Self-Regulation (Control)

Five studies [12,14,18,19,23] had a defined self-regulation (control) component. McHugh Power et al.’s (2016) offered an optional goal setting session to master new cooking skills [23]. The following two studies used goal setting to improve family mealtime dynamics. Fulkerson et al.’s (2018) participants set goals for meal planning and cooking skills development, but goal attainment did not significantly differ between intervention and control groups [18]. Overcash et al.’s (2018) goal setting session aimed to incorporate children into food preparation tasks [12]. Two studies used the self-regulation component to create health-related goals outside of cooking. White et al.’s (2019) participants set goals for healthier lifestyles, with increased cooking (*p* = 0.08) and eating (*p* = 0.08) together among adults compared to the control as results [19]. Miller et al. (2016) created SMART-R goals (specific, measurable, achievable, relevant, time-bounded, and rewarded) so participants could increase their intake of grains and protein [14], but the authors did not indicate how the rewarded component was implemented. Of the five studies that defined the self-regulation component, three revealed increased cooking confidence or self-efficacy [12,14,23], three revealed increased cooking frequency [12,14,19], and one showed null outcomes for cooking skill development [18].

#### 3.3.3. Behavioral Capability

All studies (*n* = 13) had a defined behavioral capability component [11,12,13,14,15,16,17,18,19,20,21,22,23] consisting of various lessons to improve cooking habits including, cooking method demonstrations [11,12,13,14,15,16,17,18,19,20,21,22,23], meal planning or recipe development [11,12,14,15,16,17,18,19,21,22], grocery shopping techniques [11,12,14,15,17], food storage and safety [11,12,14], MyPlate educational lessons [12,14,19], taste testing [16,20,21], mealtime strategies [18,19], and seasonal food preparations [11,13]. Three studies used the behavioral capability component to address other eating related behaviors. Pooler et al. (2017) and Overcash et al. (2018) provided food label reading lessons [12,17]. Miller et al. (2016) provided tips to minimize food waste with leftovers [14]. All studies that defined behavioral capability had positive results relating to cooking confidence [12,14,15,16,17,20,21,23] or frequency [11,12,13,14,19,22], except for two studies that also included null outcomes [18,21].

#### 3.3.4. Expectations

Only one study reported an intervention with a defined expectations component. Leone et al. (2018) launched a social media campaign to promote the benefits of shopping at the Veggie Van, a mobile produce market, with additional newsletters and nutrition demonstrations to emphasize the benefits of healthy eating [13]. As mentioned previously, Leone et al. (2018) increased frequency of fruit and vegetable consumption compared to the control group [13].

#### 3.3.5. Self-Efficacy

All studies (*n* = 13) explicitly targeted improvements in cooking-related self-efficacy [11,12,13,14,15,16,17,18,19,20,21,22,23]. To increase self-efficacy to cook or try new foods, six studies specifically included cooking demonstrations and hands-on participation to increase exposure, skills, and confidence [12,14,18,19,21,22]. McHugh Power et al. (2016) aimed to increase self-efficacy to cook and improve health behaviors by combining education, social modeling, and vicarious learning [23]. Similarly, Clifford et al. (2009) aimed to increase college students’ cooking confidence by having them watch cooking demonstrations in areas that they frequent, such as the grocery store and the kitchen [15]. Clifford et al.’s (2009) results for cooking self-efficacy increased from baseline to the 4-month follow-up for both the intervention (0.41 to 0.49) and control (0.41 to 0.49) groups [15].

Matias et al. (2021) and Leone et al. (2018) aimed to increase self-efficacy to cook whole grains or vegetables by preparing recipes with affordable and nutritious ingredients [11,13], while Brimblecombe et al. (2018) created a consumer education campaign on cooking and trying new vegetables [20]. Matias et al. (2021) results increased cooking frequency (7.0 to 10.0 times/week, *p* < 0.0001) [11]. As mentioned previously, Pooler et al. (2017) aimed to improve food resource management self-efficacy, grocery shopping techniques, and healthy eating behaviors [17]. Chapman-Novakofski and Karduck (2005) focused on improving meal preparation confidence among participants with diabetes and other outcomes related to healthy blood glucose levels, which resulted in improvements in having confidence to prepare healthful meals (*p* = 0.038) and overcome the degree of difficulty in meal preparation (*p* = 0.002) [16]. Similar to the behavioral capability outcomes, all studies that defined self-efficacy had positive results relating to cooking confidence [12,14,15,16,17,20,21,23] or frequency [11,12,13,14,19,22], except for two studies that also included null outcomes [18,21].

#### 3.3.6. Observational Learning

All studies [11,12,13,14,15,16,17,18,19,20,21,22,23] had a defined observational learning component. Most studies (*n* = 12) defined observational learning through instructor-led [11,12,13,14,15,16,17,18,19,20,21,22,23] or peer-to-peer cooking demonstrations [11]. As previously mentioned, McHugh Power et al. (2016), had participants vicariously learn by watching the instructor cook a meal and could participate if they wished [23]. Clifford et al. (2009) had a registered dietitian record a cooking demonstration and participants later watched the video during a study session [15]. Mead et al. (2013) featured stories that were broadcasted on the local radio featuring a family learning how to improve their diet and increase physical activity [21]. All studies that defined observational learning also had positive results relating to cooking confidence [12,14,15,16,17,20,21,23] or frequency [11,12,13,14,19,22], except for two studies that also included null outcomes [18,21].

#### 3.3.7. Reinforcement

Three studies had a defined reinforcement component [19,20,22]. All studies with a reinforcement component held competitions or included activities to reward participants. Brimblecombe et al. (2018) held a receipt competition for participants as a supplemental activity [20]. Roche et al. (2017) did a recipe contest for nettle (a local wild leafy green) dishes, and the winner was awarded cooking pots [22]. White et al. (2019) rewarded participants for non-cooking-related behaviors after they accomplished nutritional goals of the week [19]. Two studies that defined the reinforcement component revealed positive outcomes relating to cooking frequency [19,22] and confidence [20].

### 3.4. Related Food Outcomes

Four studies [11,13,15,20] also included results for non-cooking-related eating behaviors. Brimblecombe et al.’s (2018) affordability of vegetables (*p* = 0.004) improved and a non-significant increase was shown for fruit affordability at the end of the intervention, but not 24 weeks post-intervention [20]. Self-efficacy to increase fruit (*p* < 0.001) and vegetable (*p* = 0.001) consumption was also lower 24 weeks post-intervention than at baseline [20]. Similarly, Matias et al.’s (2021) self-efficacy to consume fruits and vegetables increased in the post-test (vs pretest; all *p* < 0.0001; effect sizes ranged 0.58–1.66), and self-reported frequency intake of fruits (−0.05 to 1.0 cups, *p* = 0.0006) and vegetables (0.00 to 1.00 cups, *p* < 0.0001) also increased [11]. Leone et al. (2018) had no significant improvements in perceived access to fruits and vegetables, but the mean difference for serving frequency between the intervention and control group at follow-up was 0.81 cups/day (*p* < 0.05) [13]. Clifford et al.’s (2009) fruit and vegetable frequency (total servings) questionnaire indicated an increase from baseline to 4-month follow-up for both the control (0.25 to 0.34) and intervention (0.29 to 0.38) groups [15].

## 4. Discussion

This narrative review evaluated how the SCT was defined in cooking interventions and which components were most prevalent in studies with the most positive outcomes. As mentioned previously, the results focused on two cooking-related outcomes, cooking self-efficacy [12,14,15,16,17,18,20,21,23] and frequency [11,12,13,14,19,21,22]. All studies included in this review were associated with various positive [11,12,13,14,15,16,17,19,20,21,22,23] or null cooking outcomes [18,21]. None of the studies in this review reported negative cooking outcomes. Among the studies with the most desirable cooking outcomes, the SCT components that were most prevalent were behavioral capability, observational learning, and self-efficacy. However, these were also the most used components in general. Studies that included null cooking outcomes defined behavioral capability, self-efficacy, observational learning, self-regulation, and environment. All included studies reported at least one positive cooking-related outcome, except for one, where no statistically significant differences in cooking skills were shown [18]. Studies that included positive cooking outcomes defined three [11,15,16], four [17,21], or five [12,13,14,19,20,22,23] SCT components. The two studies with null outcomes each defined four components [18,21].

As described in the results, none of the studies in this review defined all the SCT components. Jackson (1997) describes the application of theories in program development and evaluation to be a barrier, and that researchers often have trouble transferring theory from academic training to natural environments [24]. Given that reciprocal determinism is at the root of the SCT, interventions lacking one or more components may not sufficiently address the complex interplay between personal factors, environment, and behavior. Since all seven SCT components are closely connected to maintaining behavior change, applying fewer components will reduce the likelihood of yielding positive outcomes. In this review, the least used SCT components were self-regulation (defined five times), reinforcements (defined three times), and expectations (only defined once), and the lack of integration of these components into cooking and food skills interventions likely hinders its potential to start and sustain behavior change. The initiation of a goal may be influenced by expected outcomes (i.e., expectations), but the decision to maintain the desired behavior is influenced by people’s satisfaction of their results [25], which can be facilitated through reinforcements and self-regulation. Four of the five included studies that defined the self-regulation component included positive results for cooking frequency [12,14,19,23] and self-efficacy [12,14,23]. In a randomized controlled trial, Schnoll et al. (2001) evaluated the effectiveness of goal setting and self-monitoring to improve dietary fiber self-efficacy, which yielded increases in fiber consumption and post-intervention knowledge among the intervention group that combined both self-regulation practices [26]. In the present review, three studies that defined the reinforcement component included positive results for both cooking frequency [19,22] and self-efficacy [20]. Studies incentivized participants with cooking equipment [22], monetary compensation [19], other positive reinforcement [20] upon completing nutritional goals or winning contests. Gneezy et al. (2011) maintain that extrinsic incentives may alter behavior change or conflict with other motivations, and further hypothesized that providing monetary incentives to change behavior may help in the short run but reduce one’s intrinsic motivation long-term once incentives are removed [27].

Seven included studies that defined the environment component resulted in positive cooking outcomes [12,13,14,20,21,22,23], but Mead et al. (2013) also had a null outcome for cooking self-efficacy [21]. Similarly, most studies that defined observational learning resulted in positive cooking outcomes, except for two studies that revealed null outcomes [18,21]. In this present review, all studies that defined observational learning offered hands-on cooking demonstrations, except for one study that only used videos among participants to measure changes in cooking self-efficacy, knowledge, attitudes, and behaviors over time [15]. A cohort study by Levy et al. (2004) compared outcomes for participants either in hands-on cooking or demonstration-only classes to identify which intervention design would yield changes in attitudes, knowledge, and behaviors [28]. Their findings reveal positive cooking confidence and knowledge shifts for both groups, but the group attending hands-on cooking classes had statistically significant gains [28].

Hollywood et al. (2018) did not identify a relationship between theory-based interventions, positive outcomes, and individual behavior change technique (BCT) usage in their critical review, but they did identify an association between long-term positive behavioral change and interventions involving a practice skills element [8]. Similarly, our findings reveal the behavioral capability component is consistently associated with positive cooking confidence [12,14,15,16,17,20,23] and frequency [11,12,13,14,19,21,22]. These findings suggest increased skills development to be a key contributor to behavior change. Contrarily, one of Hollywood et al’s most common BCTs involved providing information on consequences of a behavior in general [8], but, as stated previously, expectations was the least used component in the present review. The reason for this discordance is unclear since Hollywood et al. (2018) did not connect BCT outcomes to specific studies, but it is possible that the cooking and food skills interventions that provided information on consequences were based on behavior theories other than SCT or lacked theoretical basis. Hagger et al. (2014) suggests that the CALO-RE taxonomy be used for BCT identification and application to further connect BCTs to individual theoretical components, which is necessary to test the effectiveness of an intervention [29].

A comprehensive review by Wolfson et al. (2017) suggests that further research on cooking (skills and knowledge) is needed to justify the connection between cooking and health. Their findings also suggest that intervention design often depends on assumption, but that study designs including randomization, control groups, and long-term follow-ups may develop stronger outcomes [30]. Within this review, five of the thirteen studies were randomized controlled trials [13,15,18,19,23] and six studies included a follow-up post-intervention [13,15,17,18,20,23]. Future cooking interventions should address these gaps in the literature.

## 5. Limitations and Strengths

Despite the wide scope of findings in this narrative review, limitations exist. Ten out of the 13 studies were published in the USA and Canada, which may limit generalizability to other countries. Since this narrative review only presents published original research, publication bias should be considered. Findings from this review identify gaps in current cooking interventions, which show a lack of consistent and comprehensive application of the SCT, further limiting the ability to connect this theory to interventions with positive outcomes. Although studies in this review do not explicitly define all SCT components, it is possible that they may have still implemented them. This limits the comprehensiveness of our findings, yet it is also a common issue of behavioral interventions. To further refine the SCT and its integration into cooking and food skills interventions, research studies on this topic need to clearly define how the theory is used to facilitate the reproducibility of research findings. The present literature review helps to fill this gap by elucidating the connection between individual SCT components and the most positive outcomes. Although these limitations exist, one key strength of this review is that articles were sourced from a search string, which allows included studies to be regenerated and replicated by future researchers. This study also applied specific inclusion and exclusion criteria, and only studies that defined the use of the SCT were included after abstracts were assessed.

## 6. Conclusions and Implications

There is an increasing interest in understanding the value of theory-based interventions to change cooking behaviors in adults. Findings in this review reveal behavioral capability, self-efficacy, and observational learning as the most prevalent SCT components in cooking and food skills interventions and expectations as the least common. This review confirms that cooking interventions incorporating the SCT may lead to positive results, specifically for cooking frequency and self-efficacy. However, the potential of the SCT may not be fully realized because it may not be implemented comprehensively within cooking and food skills interventions. Future studies should explicitly state how theory influences intervention design, and further connect theoretical components to key outcomes.

## Figures and Tables

**Table 1 nutrients-15-01287-t001:** Study criteria for inclusion or exclusion.

Inclusion Criteria	Exclusion Criteria
Targeted adult participants, including parents and college students and assessed their outcomes Specifically described social cognitive theory components as an influence for the intervention design Assessed cooking skills or related behaviors (e.g., meal planning) Original Research Full-text English	Only measured outcomes of children, adolescents, infants Did not report using the social cognitive theory Weight loss, Physical activity interventions without a cooking component Books, systematic literature reviews, conference abstracts Not published in English

**Table 2 nutrients-15-01287-t002:** Overview of Intervention and Study Design.

Publication County	Mode of Delivery/Intervention Provider	Target Population/Sample Size	Study Design/Duration	Intervention	Key Outcomes
Brimblecombe et al. (2018) Australia [20]	In-Person/Community leaders	Primary food shopper/*n* = 148, 85, and 73 at (T1), (T2), and (T3), respectively.	Non-controlled trial with the collection of baselines (T1), immediately before cessation of the 24-week intervention (T2) and 24-week post-intervention data (T3)/24 weeks	Price discount and consumer education-based intervention strategy that aimed to improve fruit, vegetable, and water consumption and reduce soft drink consumption	Perceived affordability of vegetables (*p* = 0.004) improved at T2 and a non-significant increase was shown for fruit at T2, but not for T3. Self-efficacy to consume more fruit (*p* < 0.001), vegetables (*p* < 0.001), and water (*p* = 0.001), and to cook and try new vegetables (*p* = 0.07) was lower at T3 compared with T1 and did not differ between T1 and T2. Self-efficacy to cook and try new vegetable was the only mediator associated with improved vegetable intake.
Chapman-Novakofski and Karduck (2005) USA [16]	In-person/Not specified	Adults with diabetes or caretakers./*n* = 239	Non-controlled trial/3 monthly 2-h classes	Community-based education program designed to improve nutrition knowledge and self-management skills among adults with diabetes.	Improvements in having confidence in changing one’s diet (*p* = 0.044), preparing healthful meals (*p* = 0.038), using the Nutrition Facts label (*p* = 0.0001), and in overcoming the degree of difficulty in meal preparation (*p* = 0.002).
Clifford et al. (2009) USA [15]	Taped Videos/Registered dietitians were filmed teaching college students cooking and grocery store activities.	Students from upper level non-health courses (18+)/*n* = 101	Randomized controlled trial with pre, post, and follow-up tests/4 weekly 15-min videos	Designed for students living off-campus to influence knowledge, attitudes, and behaviors, primarily on fruits and vegetables. SCT drove program development.	There were significant improvements in knowledge of fruit and vegetable recommendations in the intervention group compared to the control group post-intervention and at 4-month follow-up (*p* < 0.05). There were no significant changes in fruit and vegetable motivators, barriers, self-efficacy or intake.
Fulkerson et al. (2018) USA [18]	In-Person/Not specified	Main meal preparers and their children (8–12 years old)/Intervention (*n* = 81) or control (*n* = 79) groups	Randomized controlled trial/10 monthly sessions	Intervention included five parent goal setting calls and 10 monthly sessions that focused on experiential nutrition activities and education, meal planning, cooking skill development, and reducing screen time.	There were no statistically significant differences in parental meal planning and cooking skills scores between the groups.
Leone et al. (2018) USA [13]	In-Person/Not specified	Age 18 or older, English speaking, and primary food shopper/*n* = 142 participants, *n* = 111 controls	Cluster randomized controlled trial/6 months of exposure	VV (Veggie Van) is a mobile produce market selling reduced cost, locally grown produce and providing nutrition and cooking education. A social marketing campaign encouraged VV use.	No significant improvements in perceived access to fresh F&V, but participants increased their self-efficacy of adding more F&V into snacks (*p* = 0.02), making up a vegetable dish with what they had on hand (*p* = 0.03), and cooking vegetables in a way that is appealing to their family (*p* = 0.048). The intervention mean difference in F&V intake between intervention and control groups at follow-up was 0.81 cups/day (*p* < 0.05).
Matias et al. (2021) USA [11]	In-Person/Lecture led by one investigator; Cooking lab led by faculty and/or graduate student.	College Students (18+)/*n* = 171	Non-controlled trial/14 weeks (semester-long course)	Undergraduate college nutrition course (including a lab) in response to food insecurity on campus aimed to: improve attitudes, self-efficacy, and behaviors about healthful eating and cooking by implementing cooking activities and nutrition education.	Attitudes and self-efficacy scores about consuming fruits, vegetables, whole grains, and cooking were significantly higher in the post-test (vs pretest; all *p* < 0.0001; effect sizes ranged 0.58–1.66). Self-reported intake of fruits (−0.05 to 1.0 cups, *p* = 0.0006) and vegetables (0.0 to 1.0 cups, *p* < 0.0001) also increased. Cooking frequency increased (7.0 to 10.0 times/week, *p* < 0.0001), skipping meals frequency decreased (4.0 to 3.0 times/week, *p* < 0.0001), whereas no significant changes were observed for eating out, take-out, or premade meals frequency.
McHugh-Power et al. (2016) Ireland [23]	In-Person/Peer volunteers	Adults (60+ years) living alone that self-reported risk of social isolation/*n* = 100	Parallel, randomized controlled trial/8 weeks, 90 min. per week	Mealtime intervention where the volunteer and participant prepared and shared a meal. Opportunities for vicarious learning (watching the volunteer cook) and to master new cooking skills if the participant wished.	A borderline significant effect of condition over time was found for general self-efficacy as an outcome (F1, 256 = 3.578, *p* = −0.054; t256 = −1.939, *p* = 0.054; −2LL = 314.75), indicating gains in self-efficacy were greater in the in the treatment group vs control over time.
Mead et al. (2013) Canada [21]	In-Person/local community members	Primary food Preparer/*n* = 246, *n*= 133 from comparison communities	Non-randomized controlled trial/ 12 months	A community-based nutrition and lifestyle intervention aiming to improve food-related psychosocial factors and behaviors among Inuit and Inuvialuit communities	Intervention respondents increased healthy eating intentions greater than the comparison group (change 2.14, *p* < 0.0001). Intervention group significantly reduced avg. frequency of unhealthy food acquisition compared with the comparison group (change = −5.28, *p* = 0.0019). Intervention respondents acquired 4.51 fewer unhealthy foods than they did at baseline while the comparison respondents acquired 0.77 more unhealthy foods, on average. No significant improvement in food preparation scores.
Miller et al. (2016) USA [14]	In-Person/4-H program staff and extension specialists	Parent (19+) with child (9–10 years old), Primary meal preparers/Family dyads (*n* = 54)	Non-controlled trial/3 months	Designed to improve culinary skills and family mealtime through meal demonstrations, food purchasing techniques, culinary skill development, and nutrition education. Modeling mealtime quality was done through quick and easy food preparations and eating together as a family.	Adults reported increased cooking skill confidence from 75% at baseline to 86% mostly to almost always, feeling confident (*p* = 0.015). The percentage of participants reporting procuring fast food evening meals 1–2 days per week decreased from 76% to 54% (*p* = 0.033 at post-test) and those reporting no fast food evening meals increased from (16–40%). No other food preparation findings changed.
Overcash et al. (2018) USA [12]	In-person/Cooking demonstrations given by a professional chef, nutrition education lessons given by a nutrition educator, and recipe preparation under the guidance of the chef and nutrition educator	Parent child dyads (child was 9–12 years.), parent is the main food preparer, family must qualify for public assistance/*n* = 89	Non-controlled trial/6 weeks	Vegetable-focused cooking skills and nutrition education sessions aimed to improve vegetable liking, consumption, and home availability	Increased parental cooking confidence (4.0 to 4.4/5.0), healthy food preparation (3.6 to 3.9/ 5.0), child self-efficacy (14.8 to 12.4; lower score = greater self-efficacy), vegetable variety (30 to 32/37 for parent, 22 to 24/37 for child), and home vegetable availability (16 to 18/35) (all *p* < 0.05).
Pooler et al. (2017) USA [17]	In-person/nutrition and culinary educator	Low-income families/Intervention (*n*= 332); comparison group (*n* = 336).	Non-randomized controlled trial/6 weeks, once each week for 2 h	Designed to teach low-income adults how to maximize a tight budget, promote the purchase/preparation of healthier foods, and help overcome the perception that healthy foods are too expensive.	Cooking Matters participants improved scores: Use of FRM practices (*p* = 0.002) and FRM confidence (*p* < 0.001). They also worried less that food might run out before they had the resources to buy more (*p* = 0.020).
Roche et al. (2017) Ecuador [22]	In-Person/Elders, Peer leaders (guide mothers)	Mothers were selected by wealth status, number of children, community of residence, and level of participation in community events/160 participant mothers and 98 mothers in comparison communities.	Non-randomized controlled trial/24, 2-h sessions	Mothers’ cooking clubs that promoted Quichua culture and traditional foods and increased perceived self-efficacy to include two wild leafy greens.	Seventy-four per cent of mothers (*n*= 119) fed their children nettle at least once per month, versus 21% in comparison communities. Frequency ranged from once per month to daily, with a mean of 2.2 ± 1.8 times per week. Seventy per cent (*n*= 112) of mothers fed their children round-leaved dock at least once a month. The mean frequency was 1.11 ± 0.8 times per week. The likelihood of feeding children leafy greens was ~10 times greater than in comparison communities (adjusted Odds Ratio (aOR): 9.5; 95% CI: 4.37, 20.21; *p* < 0.001).
White et al. (2019) USA [19]	In-Person/extension staff	Main meal preparers and their children (9–10 years)/Intervention (*n* = 228 dyads) and Dissemination (*n* = 74)	Randomized controlled trial/Six 2-h, biweekly sessions	Out-of-school program designed to develop cooking skills and increase family mealtime dynamics.	Treatment youths increased cooking skills, (*p* = 0.03) and treatment adults increased cooking together (*p* = 0.08) and eating together (*p* = 0.08) compared with controls.

**Table 3 nutrients-15-01287-t003:** Social Cognitive Theory Components.

Publication Country	Environment	Self-Regulation (Control)	Expectations	Observational Learning	Self-Efficacy	Reinforcement
Brimblecombe et al. (2018) Australia [20]	Provide food price discounts at community stores	N/A	N/A	Cooking demonstrations	Consumer education component focused on enhancing self-efficacy to positively change intake, and to cook and try new vegetables.	Receipt competition held during one of the themed weeks
Chapman-Novakofski and Karduck (2005) USA [16]	N/A	N/A	N/A	Cooking demonstrations	Intervention aimed to improve the self-efficacy for meal preparation for diabetes, as well as other outcomes related to healthy blood glucose levels.	N/A
Clifford et al. (2009) USA [15]	N/A	N/A	N/A	Cooking Demonstrations (video)	The dietitian demonstrated quick, simple recipes, and the student guest was viewed assisting in these uncomplicated cooking tasks.	N/A
Fulkerson et al. (2018) USA [18]	N/A	Families were offered five bi-monthly goal setting sessions specific to the family	N/A	Cooking Demonstrations	Self-efficacy to identify appropriate portion sizes, increasing F&V availability in the home and cooking skills for parents and children, and promoting family mealtime environments.	N/A
Leone et al. (2018) USA [13]	Demonstrations and hands-on participation increased exposure to preparation methods and allowed for practice, which likely led to familiarity, skills, and ultimately confidence. Using different vegetables and repeated use of cooking methods promoted increased cooking confidence.	N/A	Social marketing campaigns promoted the benefits of shopping at VV. Newsletters and nutrition demonstrations addressed the benefits of healthy eating.	Cooking Demonstrations	Promoted increased self-efficacy to purchase, prepare, and eat F&V.	N/A
Matias et al. (2021) USA [11]	N/A	N/A	N/A	Instructor-led and peer-to-peer cooking demonstration	The recipes utilized affordable and nutritious ingredients, such as whole grains and vegetables, that were easy to follow to promote self-efficacy for cooking.	N/A
McHugh Power et al. (2016) Ireland [23]	Decreased social isolation during mealtime	Goal setting to master new cooking skills if the participant wished	N/A	Vicarious learning (the participant watching the volunteer cook)	Self-efficacy to cook and improve health behaviors improved by combining education, social modeling, and vicarious learning.	N/A
Mead et al. (2013) Canada [21]	Increased the availability and accessibility of healthy foods. Point-of-purchase media, such as shelf labels and posters, were displayed to help identify healthy choices.	N/A	N/A	Cooking demonstrations, radio stories featuring a family learning how to improve diet and increase physical activity.	Increased self-efficacy to engage in healthy food-related behaviors through media promotions and participation in intervention activities.	N/A
Miller et al. (2016) USA [14]	Improving family mealtime dynamics	SMART-R goals and short- and long-term goals to promote grains and proteins	N/A	Cooking demonstrations	Through family-centered activities around developing basic culinary skills, adults as well as youth may improve culinary competence, food purchasing, and mealtime behaviors.	N/A
Overcash et al. (2018) USA [12]	Providing families with bags of groceries to create the meals at home	Goal setting to incorporate children in food preparation tasks	N/A	Instructor-led cooking demonstrations	Demonstrations and hands-on participation increased exposure to preparation methods and allowed for practice, which likely led to familiarity, skills, and ultimately confidence. Using different vegetables and repeated use of cooking methods promoted increased cooking confidence.	N/A
Pooler et al. (2017) USA [17]	Providing families with take-home groceries and educational tools (recipes and materials)	N/A	N/A	Cooking demonstrations	Program components were designed to improve food resource management skill confidence, improve shopping, and healthy eating behaviors.	N/A
Roche et al. (2017) Ecuador [22]	Cooking clubs offer social support for mothers.	N/A	N/A	Cooking demonstrations	Self-efficacy to prepare traditional foods (mainly the wild leafy greens) was improved through repeated cooking and trying new recipes.	Recipe contest for nettle dishes, winner was awarded cooking pots.
White et al. (2019) USA [19]	N/A	goal setting for healthier lifestyles	N/A	Cooking demonstrations	Promoted self-efficacy by implementing the 4-H youth development approach by working in partnership with adults in experiential learning.	Rewards given for accomplishing nutritional goals of the week

## Data Availability

Not applicable.

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
