# Peer review of "Assessing the Use of Social Cognitive Theory Components in Cooking and Food Skills Interventions"

_nutrients, 2023, doi:10.3390/nu15051287_

Round 1

Reviewer 1 Report

This narrative review provided an overview of studies that applied social cognitive theory components for cooking intervention. The review is well-written, clear and easy to follow. I have some minor comments for authors to consider as follows:

L21, L147: Not all the components were used in those studies may be due to some were not relevant to the goal or less effective? Also, is it necessary to use all the SCT components? I think this would be a point of discussion for the application of SCT for different objectives.

L150-152: will this mean that these three components are the most effective?

L161: produce or product?

L164: “working” more F&V into snacks?

L163-170: Why the content related to self-efficacy was included here?

L247-250: some of this content was repeated in 3.3.1.

L280: typo in brackets.

L295: Should abbreviation – SCT be used? Please check throughout the manuscript.

L304: why using “but”?

Table 2: should the order of studies be grouped by countries and from the latest to the oldest?

Discussion: in this section, it would be interesting to discuss how the different components link/contribute to one another and eventually result in positive outcomes. I believe all the seven components are closely connected, and it won’t be possible to apply only one component to achieve positive outcomes.

Also, the efficacy of SCT components on the intervention will depend on its goal – some components may not lead to positive outcomes for certain type of intervention – it is worth having some comments around this.

Reviewer 2 Report

This review summarizes use and effectiveness of social cognitive theory components in cooking and food skills interventions. Results of this study suggest the potential for utilizing SCT when designing nutrition interventions related to cooking and food skills and highlight the importance of utilizing a theoretical framework when designing an intervention. Methods used to conduct the literature search were thorough and reproducible. Overall, the manuscript is clearly written. It would benefit from careful editing; several sections look like sentences were changed and not completely edited when revising the sentence. Also, some words used throughout, while not necessarily incorrect, are awkward. (Example: Line 20 of abstract, I would use "with" instead of "including"). A few questions/comments for clarification elsewhere are provided below.

Table 1. For inclusion and exclusion criteria, please indicate if “Used the social cognitive theory” means it was specifically indicated in the study that components were incorporated into the intervention design. Throughout the manuscript, the term “defined” is used (Lines 160, 202, 218, 229, etc.) However, in lines 384-386, this limitation makes it just a little confusing. In Line 384, do you mean the studies in this review?

Tables 2 and 3. Please provide the reference number for each study. The order of the studies doesn’t make sense but appears to be as listed in references.

Results. It would be helpful to summarize results of studies for each component such as was done in Lines 180-184. It is in some sections but not all.

Lines 392-395: I think you could add to this strength that specific inclusion and exclusion criteria were applied, and only studies that defined use of SCT were included (if that is so). As well, abstracts were reviewed for inclusion/exclusion. 

Author Response

Please the attachment.
